# Data Analysis and Filter Optimization for Pulse-Amplitude Measurement: A Case Study on High-Resolution X-ray Spectroscopy

**DOI:** 10.3390/s22134776

**Published:** 2022-06-24

**Authors:** Kasun Sameera Mannatunga, Bruno Valinoti, Werner Florian Samayoa, Maria Liz Crespo, Andres Cicuttin, Jerome Folla Kamdem, Luis Guillermo Garcia, Sergio Carrato

**Affiliations:** 1International Centre for Theoretical Physics, 34151 Trieste, Italy; ksm@sjp.ac.lk (K.S.M.); wflorian@ictp.it (W.F.S.); cicuttin@ictp.it (A.C.); jfollakamdem@gmail.com (J.F.K.); lgarcia1@ictp.it (L.G.G.); 2Department of Physics, University of Sri Jayewardenepura, Nugegada 10250, Sri Lanka; 3Department of Electronic and Telecommunications, University of Moratuwa, Moratuwa 10400, Sri Lanka; 4Instituto Nacional de Tecnologia Industrial, Buenos Aires 1650, Argentina; 5Department of Engineering, Università degli Studi di Trieste, 34127 Trieste, Italy; carrato@units.it; 6Department of Physics, University of Yaoundé I, Yaoundé, Cameroon

**Keywords:** digital signal processing (DSP), digital pulse processor (DPP), X-ray spectroscopy, silicon drift detectors, pulse-height analysis, FIR design

## Abstract

In this study, we present a procedure to optimize a set of finite impulse response filter (FIR) coefficients for digital pulse-amplitude measurement. Such an optimized filter is designed using an adapted digital penalized least mean square (DPLMS) method. The effectiveness of the procedure is demonstrated using a dataset from a case study on high-resolution X-ray spectroscopy based on single-photon detection and energy measurements. The energy resolutions of the *K*${}_{\alpha }$ and *K*${}_{\beta }$ lines of the Manganese energy spectrum have been improved by approximately 20%, compared to the reference values obtained by fitting individual photon pulses with the corresponding mathematical model.

## 1. Introduction

Single-photon counting and photon energy measurements play a central role in high-resolution X-ray spectroscopy. Many applications and research areas, such as the non-destructive analysis of cultural heritage objects, material sciences, and medical imaging, require accurate detection and energy measurement systems [1,2]. Such systems usually include a detector, amplifying stage, and filtering stage. Electrical signals coming from a detector are conditioned using a charge-sensitive amplifier (CSA). In modern systems, the analog CSA output is often immediately converted into a digital signal. Subsequently, a digital pulse processor (DPP) reshapes and filters the signal to detect and measure the amplitude of each photon pulse.

Several difficulties arise when performing single-photon detection and energy measurements owing to the presence of noise, unstable signal backgrounds, and spurious pulses. A significant source of noise is the detector itself, as it is at the beginning of the amplification chain. Although an important noise reduction can be obtained by cooling the detector [3,4], it is still crucial to perform optimal filtering to improve the signal-to-noise ratio (SNR) for high-resolution spectroscopy [5]. Focusing on the latter strategy, an optimal DPP must also be able to efficiently deal with high pulse rates.

The central element of a DPP is a digital shaping filter, usually a finite impulse response (FIR) filter. In this paper, we present an FIR optimization procedure based on the digital penalized least mean square (DPLMS) method [6]. The procedure requires an accurate mathematical model of the input pulse and appropriate characterization of the noise. By analyzing an experimental dataset, it is possible to separate the pulse signal and the noise of the system. We also define a set of constraints to optimize the filter coefficients by improving the SNR as well as providing a flat-top at the output pulse where its peak value corresponds to the true amplitude of the input pulse.

We apply the DPLMS-based optimization procedure to a case study on high-resolution digital X-ray spectroscopy and provide a comparison with other related methods. We consider a simple trapezoidal filter as the starting point. Successively, we modify this filter to reject the error introduced by a constant background slope. Finally, we design an optimal filter that takes into account (i) the characteristic shape of the pulse and (ii) the specific noise. Each of these filters is implemented using a software-simulated DPP to obtain the corresponding energy spectra from an experimental dataset.

The next sections of this paper are organized as follows: Section 2 introduces the literature review that lays the foundation for the proposed methodology. Section 3 briefly describes a typical X-ray spectroscopy detection system and the experimental data on which our case study is based. Section 4 presents two FIR filter design methods for pulse-amplitude measurement. Section 5 describes the data analysis to obtain the information needed by the DPLMS-based optimization procedure, including the ideal pulse modeling and noise characterization. A brief comparison of the described methods is presented in Section 6. Section 7 describes an additional statistical analysis of the experimental data which uncovers some weaknesses of the system, providing an important indication to improve the spectroscopy system. Finally, Section 8 presents the conclusions of this study.

## 2. Related Works

Several digital pulse-processing techniques have been introduced for spectral measurements over the years. In 1993, Jordanov and Knoll [7] presented a real-time DPP using a moving average technique built on high-speed programmable logic devices (PLD) and fast-TTL integrated circuits. This implementation included a conventional quasi-Gaussian analog shaper after the CSA. In 1994, they extended their work by introducing fast recursive digital algorithms implemented on a personal computer (PC) for the synthesis of symmetric triangular and trapezoidal pulse shapes, thereby replacing the traditional analog pulse shapers [8]. Later that year, Jordanov V. et al. [9] implemented digital shaper algorithms on dedicated hardware by avoiding the use of a PC for offline pulse processing.

Guzik Z. and Krakowski T. [10] presented a full set of recursive algorithms based on the Z-transform for trapezoidal pulse shaping with pole-zero cancellation for exponentially decaying input pulses. The complete system was implemented on an FPGA and included energy reconstruction, baseline restoration, trigger generation, and event acceptance. The use of this approach is limited because of the complexity of deriving recursive formulas for different input pulse shapes generated by various pulse detection systems.

Sajedi S. et al. [5] proposed an FPGA-based non-linear recursive filter design for high-rate pulse feature extraction in nuclear medicine imaging and spectroscopy. Real data were obtained directly from the pre-amplifier of the detection system. It was then fitted offline using the least-squares curve-fitting method on a PC to obtain the deterministic pulse model. The pulse-shape model was then used to generate look-up tables (LUT) and implement non-linear recursive filters. The main disadvantage of this system is the large usage of memory elements. The aforementioned methods disregard the noise present in the system.

In contrast to the progression of recursive IIR methods, non-recursive FIR-based digital signal-processing methods have been independently developed for pulse-height analysis. In 1996, Gatti E. et al. [11] introduced a method for calculating the FIR filter coefficients for nuclear spectroscopy with time-domain constraints and the uncorrelated noise present in the signal. The filter was obtained by solving a set of linear equations derived by expressing the filter shape and equivalent noise charge as a modified Fourier sine series. Later, Gatti E. et al. [12] modified the method and incorporated experimental noise by estimating the noise power spectral density of data obtained from an analog shaper in the absence of pulses.

In 2002, Riboldi et al. proposed a numerical approach based on the least mean squares method [13] to calculate the optimum FIR filter coefficients. In 2004, Gatti E. et al. presented the fully formalized method [6], named the DPLMS. A drawback of this method is that it directly estimates noise using the sampled data stream by assuming that no correlated noise is present. In 2007, Riboldi S. et al. [14] extended the DPLMS method by addressing the correlated noise. Additionally, there are several publications that detailed FIR-based digital pulse-shaping systems by utilizing the DPLMS method implemented on FPGAs [15,16,17] and SoC-FPGAs [18].

Considering the mentioned contributions, it can be noted that the DPLMS optimization method improves the SNR of the output pulse. However, the application of this method requires the knowledge of the real characteristics of the noise and an accurate mathematical model for the noiseless pulse. We present an effective procedure to evaluate the model, characterize the noise present in the system, and include this information along with other constraints in the DPLMS method.

## 3. X-ray Spectroscopy Detection System

Particle detectors are central devices in X-ray spectroscopy. They are available in different technologies, such as gaseous ionization detectors, silicon drift detectors (SDD), photodiode detectors, and photomultipliers. Among these, interest in SDDs for single-photon detection has been constantly growing since its introduction by Gatti and Rehak in 1983 [19,20]. Owing to their intrinsic low noise and ability to operate with high photon rates, they are widely used in X-ray spectroscopy.

X-ray photon detectors generate a small amount of electric charge for each absorbed photon. This charge is proportional to the energy of the photon and produces a very small and short current pulse, which typically requires amplification and filtering before the analysis. The first amplification stage is commonly performed using a CSA that integrates a small charge, producing a relatively large voltage step [21,22]. This voltage step is further amplified and filtered using a pulse-shaping amplifier (PSA), which produces a semi-Gaussian pulse ready for digitization.

A typical single-photon detection system in X-ray spectroscopy consists of a detector, CSA, PSA, analog-to-digital converter (ADC), and DPP for pulse-amplitude measurement, as shown in Figure 1.

In these photon detection systems, the major electronic noise contributors are the CSA and the leakage current of the detector. The first CSA was proposed in 1956 by Gatti [23]. Subsequently, continuous modifications have been made to improve the SNR [24,25].

An idealized noiseless CSA output pulse can be described by an exponential upward step-like pulse, expressed as follows:(1)$$V\left(t\right)=\left\{\begin{array}{cc} 0, & t⩽t_{0};\\ A(1-e^{\frac{-(t-t_{0})}{\tau }}), & t>t_{0}; \end{array}\right.$$
where $t_{0}$ is the pulse arrival time and τ is the exponential rise time of the CSA, which is limited and determined by its slew rate and the non-zero charge integration time.

The CSA integrates not only the charge produced by the absorbed photons but also the leakage current of the detector. Owing to this small constant leakage current, the CSA produces a ramp with a constant slope. The complete output signal of the CSA can be modeled as a superposition of the ramp, ideal pulse, and noise, as follows:(2)$$V\left(t\right)=\left\{\begin{array}{cc} B_{0}+B_{1}t+n\left(t\right), & t⩽t_{0};\\ A\left(1-e^{\frac{-(t-t_{0})}{\tau }}\right)+B_{0}+B_{1}t+n\left(t\right), & t>t_{0}; \end{array}\right.$$
where $B_{0}$ denotes an arbitrary offset, $B_{1}$ denotes the angular coefficient corresponding to the constant slope of the baseline ramp, and n(t) is the noise component. For digitized signals, the CSA output can be rewritten by replacing the continuous-time variable *t* with the discrete index *i*, which expresses time in units of sampling periods, as follows:(3)$$x_{i}=\left\{\begin{array}{cc} B_{0}+B_{1}i+n_{i}, & i⩽t_{0};\\ A\left(1-e^{\frac{-(i-t_{0})}{\tau }}\right)+B_{0}+B_{1}i+n_{i}, & i>t_{0}; \end{array}\right.$$

The parameters *A* and $B_{0}$ are then expressed in ADC value, $t_{0}$ and τ are expressed in units of sampling periods, and $B_{1}$ in ADC value per sampling period.

### Experimental Data

In this paper, we consider an experimental dataset from a typical X-ray fluorescence experiment. The data were obtained by digitizing the signal from a low-noise CSA coupled with a SDD-based single-photon detection system [24,26,27,28] without a PSA. The dataset contains 2929 segments sampled at 40 Mhz with a 12-bit ADC. Each segment is 512 samples long and contains a single-photon pulse, as shown in Figure 2. This dataset was taken under normal operating conditions; thus, it includes real noise. A trigger system and a circular buffer allowed capturing the traces with the pulses starting around the 200th sample.

One characteristic of this dataset is that all photons present a different offset value. This was caused by the background slope and the random arrival times of the photons [29,30].

## 4. Digital Pulse Shaping

In traditional pulse-processing systems, the output of the CSA goes through a shaping stage, which improves the SNR and converts step-like pulses to pulses suitable for subsequent digital acquisition and signal processing.

A typical $CR-{\left(RC\right)}^{n}$ analog pulse-shaper amplifier consists of one differentiator followed by *n* integrators to produce a semi-Gaussian output pulse. This type of analog shaper can be replaced by modern digital shaping systems, which offer several advantages [31,32]. In these systems, the CSA output is directly digitized using a fast ADC and is immediately processed by a customized DPP. The pulse shaping can be digitally implemented in a more controlled way than with an analog circuit. An ideal DPP produces the most accurate and precise amplitude measurement of the CSA output.

A simplified block diagram of a DPP [33] for high-resolution X-ray spectroscopy is shown in Figure 3. It consists of two separate data channels: one for the precise detection of the photon arrival and the other one for shaping the input pulse based on an FIR filter. The pulse detection module detects the arrival time and decides when to retrieve the pulse amplitude. The module also controls a FIFO to store the amplitude from the digital shaping filter output at the correct sampling time. The FIFO allows asynchronous storage of the amplitudes of pulses, which typically occurs at random times, and a synchronous regular reading by the system hosting the DPP.

The digital shaping filter should fulfill the following requisites:Be independent of any offset.Be independent of any constant background slope.Optimize the SNR, according to real noise characteristics.Generate a flat-top to mitigate the uncertainty of the pulse arrival-time detection.

An additional requirement regards the time resolution of the filter, which strongly depends on its length. If two photons are separated by less than the filter integration time, the filter may not be able to properly process each one. For high photon-rate regimes, it is essential to make the shortest possible filter without significantly sacrificing the filtering capabilities. Taking this aspect into account, the length of the filters considered in this paper has been fixed to 80 taps at 40 Msps.

### 4.1. Trapezoidal FIR Filter

As a starting point, a trapezoidal output filter approximation is implemented to measure the pulse amplitude [34]. Figure 4 shows the filter coefficients and the output pulse corresponding to an experimental input pulse, such as that in Figure 2.

The underlying idea of this filter is that the amplitude of the pulse can be calculated by waiting for the CSA output to settle within an acceptable error and then subtract from it the baseline before the arrival of the pulse. To attenuate the white noise, a simple average before and after the pulse allows for a more precise estimation of the pulse amplitude. The number of positive, null, and negative filter coefficients, respectively, correspond to the parameters $t_{R}$, $t_{FT}$, and $t_{F}$. The $t_{R}$ positive coefficients of the filter compute a moving average and determine the rise time of the output pulse. Their value is constant and equal to $1/t_{R}$. The $t_{FT}$ central null coefficients define the time waited for the output pulse to settle within an acceptable error and the duration of a nearly flat-top of the output pulse. Finally, the $t_{F}$ negative coefficients compute another moving average and determine the fall time of the output pulse. Their value is constant and equal to $1/t_{F}$. These three parameters are bounded by the condition $t_{R}+t_{FT}+t_{F}=80$, because the considered length of the filter has been fixed at 80. Regarding the FIR filter output, it can be seen that there is some top flatness that could reduce the error in the amplitude measurement. However, it can also be observed that the output of the filter presents an offset that introduces an error in the amplitude measurement. This error increases with the background slope, but it can be corrected by modifying the FIR filter, as explained in the next subsection.

### 4.2. Geometrically Derived FIR Filter

To correct the abovementioned error due to the background slope, we perform a study on the geometry of the pulse. A typical photon pulse with its geometrical features is shown in Figure 5. The height of the two points in the middle of the segments indicated with $t_{R}$ and $t_{F}$ correspond to the average height computed over those segments. The trapezoidal filter computes the difference between these average values as an estimation of the pulse amplitude *A*, but we can see that this difference is A+D instead of the expected true value *A*. 

The amplitude error *D* due to the background slope is related to tanα, as follows:(4)$$tan\;\alpha =\frac{D}{\frac{1}{2}t_{R}+t_{FT}+\frac{1}{2}t_{F}}$$

The value of tanα can be estimated using the least-squares method considering the $t_{R}$ samples before the arrival of the pulse. A closed-form expression for estimating tanα can be written as follows (see Appendix B for details):(5)$$tan\;\alpha \approx \sum_{i=0}^{t_{R}-1}-6\left(\frac{1+t_{R}-2i}{t_{R}^{3}-t_{R}}\right)x_{i}$$

From Equations (Equation 4) and (Equation 5), the error *D* is estimated as
(6)$$D=\sum_{i=0}^{t_{R}-1}-6\left(\frac{1+t_{R}-2i}{t_{R}^{3}-t_{R}}\right)\left(\frac{1}{2}t_{R}+t_{FT}+\frac{1}{2}t_{F}\right)x_{i}$$
and the correct amplitude *A* is then calculated as follows:(7)$$A=\frac{1}{t_{F}}\sum_{i=t_{R}+t_{FT}}^{t_{R}+t_{FT}+t_{F}-1}x_{i}\;-\frac{1}{t_{R}}\sum_{i=0}^{t_{R}-1}x_{i}\;-\sum_{i=0}^{t_{R}-1}-6\left(\frac{1+t_{R}-2i}{t_{R}^{3}-t_{R}}\right)\left(\frac{1}{2}t_{R}+t_{FT}+\frac{1}{2}t_{F}\right)x_{i}$$

By simplifying and rearranging the previous expression, we can show that the amplitude can be computed as a linear combination of the sequential data $x_{i}$ with constant coefficients:(8)$$A=\sum_{i=t_{R}+t_{FT}}^{t_{R}+t_{FT}+t_{F}-1}\frac{1}{t_{F}}x_{i}+\sum_{i=0}^{t_{R}-1}\left[-\frac{1}{t_{R}}+6\left(\frac{1+t_{R}-2i}{t_{R}^{3}-t_{R}}\right)\left(\frac{1}{2}t_{R}+t_{FT}+\frac{1}{2}t_{F}\right)\right]x_{i}$$

It can be seen that the pulse amplitude *A* can be continuously evaluated by an FIR filter whose coefficients are described as follows:(9)$$c_{i}=\left\{\begin{array}{cc} \frac{1}{t_{F}}, & 0⩽i<t_{F};\\ 0, & t_{F}⩽i<t_{F}+t_{FT};\\ -\frac{1}{t_{R}}+6\left(\frac{1+t_{R}-2i}{t_{R}^{3}-t_{R}}\right)\left(\frac{1}{2}t_{R}+t_{FT}+\frac{1}{2}t_{F}\right), & t_{F}+t_{FT}⩽i<t_{F}+t_{FT}+t_{R}; \end{array}\right.$$

The central null coefficients determine the nearly flat-top region of the output pulse. Figure 6 shows these *geometrically derived* (GD) FIR coefficients with the parameters $t_{R}=35$, $t_{FT}=10$, and $t_{F}=35$, and the output pulse obtained with this filter is applied to an experimental pulse. As expected, this GD FIR filter suppresses the offset and background slope of the input pulse.

## 5. Data Analysis and FIR Filter Optimization

As described in Section 4, it is desirable that the shaping filter output pulse has the highest possible SNR and a flat-top to mitigate the uncertainty of the photon arrival time. These two main conditions directly contribute to achieving an optimal energy resolution [35]. To satisfy these conditions, based on the analysis of the experimental data, we define an accurate mathematical model for the input pulse (Section 5.1) and characterize the noise (Section 5.2).

The input pulse model is essential because (i) it allows the noise characterization by correctly separating the stochastic component (noise) from the deterministic signal and (ii) it contributes to a correct calculation of the FIR filter coefficients to determine a flat-top at the output.

The adapted DPLMS method, based on the pulse model and the characterized noise, is presented in Section 5.3.

### 5.1. Pulse Modeling

The model parameters *A*, $B_{0}$, $B_{1}$, $t_{0}$, and τ of the deterministic noiseless input pulse described in Equation (Equation 3) are estimated numerically by fitting the model to the experimental data. A typical experimental pulse with a fitted model and the corresponding residuals are shown in Figure 7. The residuals would correspond to the stochastic component and should be considered as the noise {$n_{i}$}.

The residuals plot in Figure 7 shows a relatively large spike around the starting point of the pulse, which indicates inaccurate modeling. Therefore, we propose a bi-exponential pulse model with the same number of parameters, described by Equation (Equation 10).
(10)$$x_{i}=\left\{\begin{array}{cc} B_{0}+iB_{1}+n_{i}, & i⩽t_{0};\\ A\left(1-2e^{\frac{-(i-t_{0})}{\tau }}+e^{\frac{-2(i-t_{0})}{\tau }}\right)+B_{0}+iB_{1}+n_{i}, & i>t_{0}; \end{array}\right.$$

This model is a heuristic model that can be analytically derived from some assumptions about the transfer function of the CSA (see Appendix A for details). The result of the fitting with the bi-exponential model is shown in Figure 8. The improvement can be observed in the residuals, which do not present evident artifacts.

Table 1 shows a comparison of both models using the mean quadratic residuals, peak-to-peak residuals, and Akaike information criterion [36] evaluated over all fitted pulses. The calculated values of these indicators confirm that the bi-exponential model is significantly more accurate than the exponential model.

Figure 9 shows the distributions of all fitted model parameters when using the bi-exponential model. The average values of the fitting parameters along with their standard deviations are presented in Table 2. The parameter $B_{0}$ is a vertical offset that randomly changes from pulse to pulse and is distributed rather uniformly. The arrival time $t_{0}$ and the slope coefficient $B_{1}$ are also stochastic variables that change from pulse to pulse, but they closely follow Gaussian distributions. In contrast, the mean value of τ is the estimate of the only parameter that characterizes the ideal pulse shape, and it is assumed to be equal for all photons.

Because the amplitude of a photon pulse is proportional to the photon energy, the histogram of the fitted amplitudes in Figure 9a represents the energy spectrum of the detected photons that, in this study, corresponds to transition lines of Manganese (Mn). The two main peaks correspond to the lines $K_{\alpha }$ and $K_{\beta }$, respectively, at 5890 eV and 6490 eV [37], and the third small peak (around 140) corresponds to 90-degree Compton-scattered photons.

### 5.2. FIR Input Noise Characterization and Output Noise Estimation

Based on the fitting of the pulses and the residuals calculation, we can proceed to define the noise at the output of the filter. Let *y* be the convolution of an input signal *x* with a *k*-tap FIR filter,
(11)$$y_{j}=\sum_{i=0}^{k-1}c_{i}\;x_{j-i}$$

Assuming that *x* is the noise at the input, a statistical description of the noise at the output *y* is needed. Hence, the variance of *y*, denoted by $\sigma_{y}^{2}$, can be written as
(12)$$\sigma_{y}^{2}=\langle {\left(y-\langle y\rangle \right)}^{2}\rangle =\sum_{i=0}^{n-1}\sum_{j=0}^{n-1}c_{i}\;c_{j}\;\underset{\mathrm{Covariance}\;\mathrm{Matrix}\;V_{i,j}}{\underset{︸}{\langle x_{i}-\langle x_{i}\rangle \rangle \;\langle x_{j}-\langle x_{j}\rangle \rangle }}$$

In this case, the noise in the experimental data is considered stationary. Moreover, the autocovariance matrix becomes the normalized autocorrelation function (ACF) when the data $\left\{x_{i}\right\}$ are standardized such that the mean $\langle x_{i}\rangle$ is 0 and the standard deviation $\sigma_{x}$ is 1; in this case, Equation (Equation 12) can be rewritten in terms of the ACF as
(13)$$\sigma_{y}^{2}=\sum_{i=0}^{k-1}\sum_{j=0}^{k-1}c_{i}\;c_{j}\;ACF(|i-j|)$$
where the normalized ACF is estimated from the experimental data $\left\{x_{i}\right\}$ as follows:(14)$$ACF\left(j\right)=\frac{\sum_{i=1}^{N-j}x_{i}\;x_{i+j}}{\sum_{i=1}^{N-j}x_{i}^{2}}$$

Here, *N* is the maximum number of consecutive samples available in the residuals. From the experimental dataset, the normalized ACF of each segment was calculated using Equation (Equation 14) where N=512 and $\left\{x_{i}\right\}$ are the model-fitting residuals. Then, all ACFs estimated on each segment were averaged to be later used in Equation (Equation 13). Figure 10 shows the first 80 values of the normalized average ACF.

The normalized average autocorrelation function in Figure 10 shows an abrupt change, from 1 with lag=0 to about 0.7 with lag=1, and from there it follows a smooth decay to slightly negative values from lag=40 onward. The first abrupt change would correspond to a white noise component, whereas the smooth decay would correspond to relatively low-frequency components of the noise spectrum.

### 5.3. Adapted DPLMS Filter Optimization

The original DPLMS method considers a number of constraints and a set of corresponding weights. These constraints define the objectives of the optimization, and their corresponding weights determine their relative relevance. In this way, it is possible to reach a trade-off among goals that cannot all be fully satisfied. We adapt these constraints, taking into account the characteristics of the experimental system. Based on the optimum digital shaping filter requisites, described in Section 4, four constraints are defined. One constraint removes the constant offset in the signal. Another one removes the background ramp due to the leakage current. The SNR is maximized using a constraint that minimizes the variance at the output of the filter in the presence of noise, as described in Equation (Equation 13). Finally, the flat-top is determined by a constraint that minimizes the error between the amplitude of an ideal input pulse and the convolution of the filter with that input pulse model.

The weights of each constraint can be arbitrarily set between zero and infinite. By adjusting the relative values of the different weights, it is possible to obtain diverse trade-offs among competing requirements.

To be immune to the offset introduced by the term $B_{0}$ of the pulse model in Equation (Equation 10), it is enough that the *k*-tap FIR filter coefficients $\left\{c_{i}\right\}$ comply with the following constraint:(15)$$\sum_{i=0}^{k-1}c_{i}=0$$

The ramp slope given by the angular coefficient $B_{1}$ will also introduce a bias in the pulse-amplitude measurement. To cancel this effect, we impose the following constraint:(16)$$\sum_{i=0}^{k-1}c_{i}\;i=0$$

Another source of error when measuring the amplitude of the pulse is the pulse arrival-time detection method. Because the value of the pulse amplitude is captured at the output of the FIR filter after a fixed time from the pulse detection, any error in the determination of the pulse arrival time will be automatically transferred to the sampling time of the filter output. By holding the amplitude value for a determined time, a flat-top is created in the output pulse, effectively compensating for the error in the estimation of the photon arrival time. This condition is expressed by the following constraints on every output $y_{j}$ of the flat-top region,
(17)$$y_{j}=\sum_{i=0}^{k-1}c_{i}\;x_{j-i}=A,\;\;\;\;\;\;\;j\in \left[t_{R},t_{R}+t_{FT}-1\right]$$
where *x* is a pulse modeled with Equation (Equation 10) without noise, and *A* is its amplitude.

To improve the amplitude measurement, the noise in the filter output needs to be minimized. This can be achieved by decreasing the filter output variance described by Equation (Equation 13). By considering this requirement, and the constraints expressed in Equations (Equation 15)–(Equation 17), we define the following quadratic cost function:(18)$$\begin{array}{c} \Psi \left(c_{0},c_{1},\ldots ,c_{k-1}\right)=\alpha_{1}{\left(\sum_{i=0}^{k-1}c_{i}\right)}^{2}+\alpha_{2}{\left(\sum_{i=0}^{k-1}c_{i}\;i\right)}^{2}+\alpha_{3}\sum_{j=t_{R}}^{t_{R}+t_{FT}-1}{\left(\sum_{i=0}^{k-1}c_{i}\;x_{k+j-i}-A\right)}^{2}+\\ \alpha_{4}\sum_{i=0}^{k-1}\sum_{j=0}^{k-1}c_{i}\;c_{j}\;ACF_{\left|i-j\right|} \end{array}$$

The significance of each constraint is determined by the relative values of the weights $\left\{\alpha_{j}\right\}$ associated with each corresponding quadratic term. By minimizing the function Ψ for a given set of weights $\left\{\alpha_{j}\right\}$, it is possible to obtain an optimal set of coefficients ${\left\{c_{i}\right\}}_{opt}$, that is,
(19)$${\left\{c_{0},c_{1},\ldots ,c_{k-1}\right\}}_{opt}=\underset{\left\{c_{0},c_{1},\ldots ,c_{k-1}\right\}}{argmin}\Psi \left(c_{0},c_{1},\ldots ,c_{k-1}\right)$$

If a weight $\alpha_{i}$ is set to zero, then the associated constraint is completely ignored, whereas in the limit where the weight approaches infinite, the constraint tends to be fully satisfied by the optimization, as in the case of the Lagrange’s multipliers method. There are no formulated rules to obtain the best weights $\left\{\alpha_{j}\right\}$, and these are manually adjusted after multiple trials.

An optimized set of coefficients was generated by minimizing the quadratic cost function Ψ (Equation (Equation 18)). The minimization was performed using numerical software optimization routines, where the function Ψ was set by carefully selecting the values of the weights $\left\{\alpha_{i}\right\}$. Figure 11 shows the generated 80-tap FIR filter and the filtered output pulse corresponding to an experimental input pulse.

## 6. Comparison of the Described Methods

We have described and applied four methods to obtain the energy spectrum from the same experimental dataset of single-photon pulses. One method consists of fitting each single-pulse trace with a model where the amplitude is one of the fitting parameters. The other three methods are based on FIR filtering for pulse-amplitude measurements. In all cases, the histogram of the amplitudes estimates the energy spectrum of the detected photons. Each histogram has been approximated using a weighted sum of three Gaussian distributions. The two largest peaks correspond to X-ray fluorescent photons, and the smallest one to Compton-scattered photons (see histogram of amplitudes in Figure 9). Given that the two main peaks correspond to the $K_{\alpha }$ and $K_{\beta }$ transition lines of Mn, whose energies are, respectively, 5890 eV and 6490 eV, it is possible to calibrate the system [37], establishing a linear correspondence between the amplitude expressed in ADC channels and the energy expressed in eV.

Table 3 shows the full width at half maximum (FWHM) obtained with each method with its corresponding uncertainty. The FWHM for the 90-degree Compton-scattered photons is not considered due to insufficient statistical representation in the dataset. The background slope introduces an offset error in the measured amplitude. This error is corrected in the GD FIR and fitting methods, allowing a simple one-point calibration and making the filter immune to possible slope variations after calibration. On the other hand, the trapezoidal FIR method requires a two-point calibration process to correct the background slope error. The DPLMS FIR results have been achieved by emphasizing the energy resolution, placing the slope-error correction in a lower priority.

The best results in terms of energy resolution have been obtained with an FIR filter optimized with the adapted DPLMS method. This method is the only one that considers the specific noise in the dataset and simultaneously allows control of the top flatness of the output pulse. In order to obtain the best results, we have relaxed the weight of the slope-error correction.

The fitting of individual photon pulses is a numerically heavy procedure to obtain the pulse amplitude. Although this method is not suitable for online data processing, it is expected to provide the most precise spectrum. The best results, however, were obtained with the DPLMS FIR method which outperformed the fitting procedure by about 20% in terms of energy resolution.

## 7. System Statistical Analysis

Besides the noise, there might be other possible imperfections in the system that could contribute to the degradation of the final energy resolution. In this section, we focus our attention on two important defects: (i) the arrival-time detection error and (ii) the non-linearity of analog amplification. These imperfections can be revealed through a statistical analysis of the experimental data. The first step of the analysis consists of fitting the pulse model to each photon trace. The fitting is performed by adjusting the model parameters to minimize the mean square error of the residuals. Thus, each trace will generally have a different set of parameters. For a given model parameter, the ordered set of obtained values constitutes a vector. In the second step, possible correlations among fitted parameters are investigated using the correlation distance $C_{D}\left(x,y\right)$ between two vectors x and y, defined as follows:(20)$$C_{D}\left(x,y\right)=1-\frac{\sum_{i=1}^{n}\left(x_{i}-\bar{x}\right)\left(y_{i}-\bar{y}\right)}{\sqrt{\sum_{i=1}^{n}{\left(x_{i}-\bar{x}\right)}^{2}\sum_{i=1}^{n}{\left(y_{i}-\bar{y}\right)}^{2}}}$$
where $x_{i}$ and $y_{i}$ are any two adjusted parameters of the fitting model used to approximate the *i-th* trace of an acquired photon, $\bar{x}$ and $\bar{y}$ are the mean values of these parameters computed over all *n* analyzed photon traces. The value of $C_{D}\left(x,y\right)$ is interpreted as follows: 

$C_{D}=1$ ⇒ x,y are uncorrelated vectors

$0⩽C_{D}<1$ ⇒ x,y are positively correlated

$C_{D}>1$⇒x,y are negatively correlated

Table 4 shows the correlation distance of all possible pairs among the five fitting parameters $\left\{A,t_{0},\tau ,B_{0},B_{1}\right\}$ of the bi-exponential model described in Equation (Equation 10). The table shows that the system has (i) a positive correlation between amplitudes *A* and arrival times $t_{0}$, (ii) a negative correlation between exponential times τ and arrival times $t_{0}$, and (iii) a negative correlation between offsets $B_{0}$ and slope coefficients $B_{1}$.

The significant correlations of the arrival times with the amplitudes and with the exponential times are due to the photon pulse detection method used to capture the experimental trace. These correlations are also shown in the scatter plots in Figure 12 and Figure 13. This finding reveals a weakness in the pulse detection method because an ideal detection is supposed to give the same arrival time independently of the amplitude and shape of the input pulse.

Even though we considered that the background slope is constant, a negative trend can be observed between offsets and slopes in Figure 14. This effect is also reflected in the value of the correlation distance between the offset and slope vectors. This value indicates that the gain of the CSA is not constant and reveals a non-linear behavior which in turn determines an error in the arrival-time detection.

By further elaborating the relationship between the amplitude of the photons and their arrival time using a clustering procedure, three clear clusters were found and are shown in Figure 15.

The two largest clusters (left and center) correspond to the $K_{\alpha }$ and the $K_{\beta }$ transition lines of the Mn, while the third and smallest cluster (right) can be identified as Compton-scattered photons. From the linear least-squares fit of the centroids of the clusters (large red points), it is found that the arrival time $t_{0}$ depends approximately on the amplitude *A* according to $t_{0}=189.967+0.0395A$.

## 8. Conclusions

In this study, we have presented a procedure to obtain a set of optimized FIR filter coefficients for the pulse-amplitude measurement targeted for high-energy-resolution X-ray spectroscopy. This includes (i) the extraction of the necessary information through the statistical analysis of the available experimental data, (ii) accurate mathematical pulse modeling and noise characterization, (iii) the formulation of constraints according to different requirements, and (iv) the numerical optimization of the digital filter.

We have used experimental step-like pulses from the output of a low-noise CSA in an SDD-based single-photon detection system to illustrate the proposed procedure. The constraints for rejecting the offset, background ramp, and improving the SNR while creating a flat-top at the output pulse have been described in this study.

The FIR filter coefficients generated with the DPLMS-based optimization procedure have shown an improvement of about 20% in the energy resolution of both the *K*${}_{\alpha }$ and *K*${}_{\beta }$ lines of the Mn energy spectrum with respect to the spectrum derived from the distribution of amplitudes obtained with least-squares fitting of individual photon pulses. The improvement is also noticeable when comparing the energy resolution obtained with the other presented FIR-based methods. The optimal energy resolution achieved through the adapted DPLMS method is mainly due to the fact that this method considers the specific noise in the acquired raw data and allows the control of the output pulse top flatness.

Two unexpected anomalies of the pulse detection system and associated front-end electronics have been observed after analyzing the experimental data by means of scattered plots and correlation distances between pulse model parameters. A non-linearity of the CSA has been indirectly uncovered by observing a negative correlation between the offset and the baseline slope. A significant dependence on the detected arrival time of the pulse amplitude has also been observed, exposing an imperfection in the pulse detection method.

These statistical results provide important indications for possible improvements of the spectroscopy system.

## Figures and Tables

**Figure 1 sensors-22-04776-f001:**
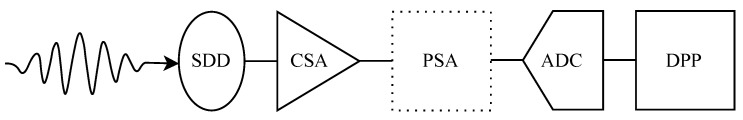
Block diagram of a typical single-photon detection system showing the incident photon on the silicon drift detector (SDD), charge-sensitive amplifier (CSA), optional pulse-shaping amplifier (PSA), analog-to-digital converter (ADC), and digital pulse processor (DPP).

**Figure 2 sensors-22-04776-f002:**
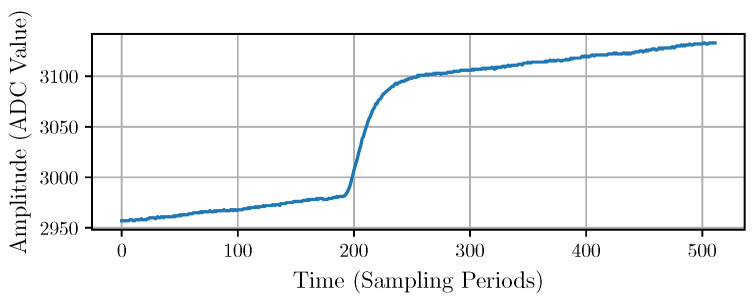
Typical experimental single-photon pulse at the output of the charge-sensitive amplifier (CSA).

**Figure 3 sensors-22-04776-f003:**
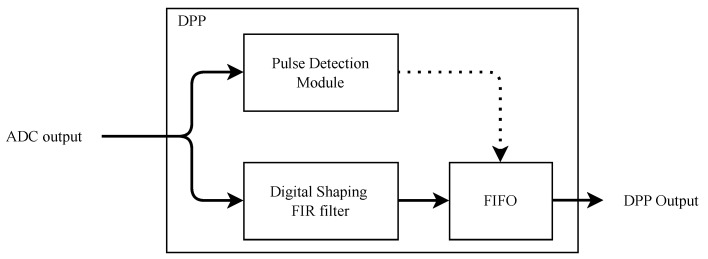
Simplified block diagram of the digital pulse-processing unit.

**Figure 4 sensors-22-04776-f004:**
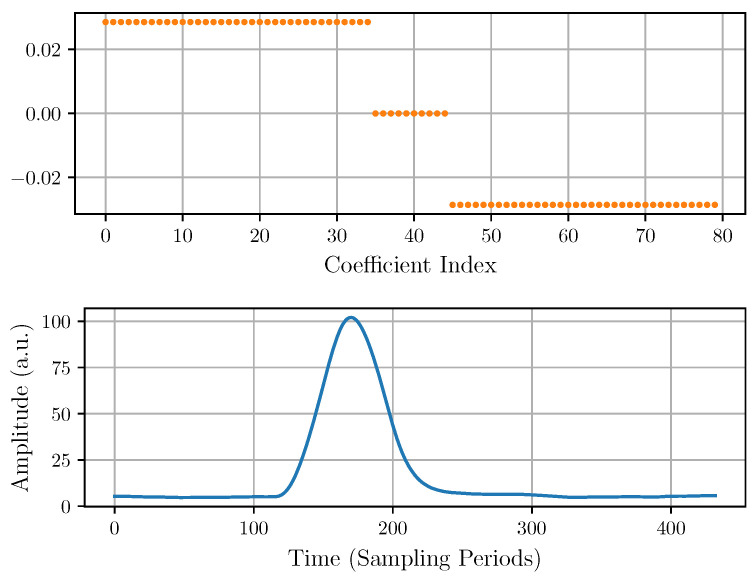
Trapezoidal FIR coefficients (**top**) and the output pulse (**bottom**) corresponding to an experimental input pulse like that of Figure 2.

**Figure 5 sensors-22-04776-f005:**
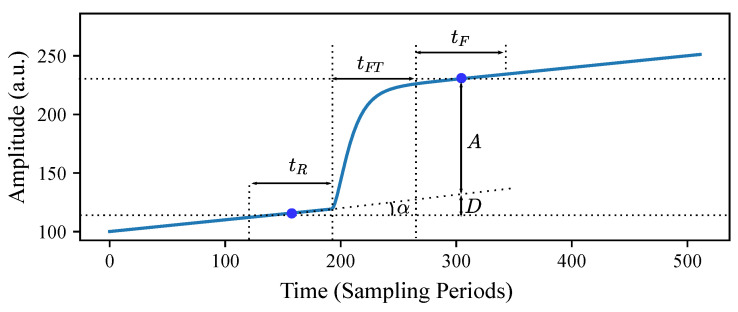
Typical photon pulse with its geometrical features highlighted. The two points in the middle of $t_{R}$ and $t_{F}$ segments correspond to their average values.

**Figure 6 sensors-22-04776-f006:**
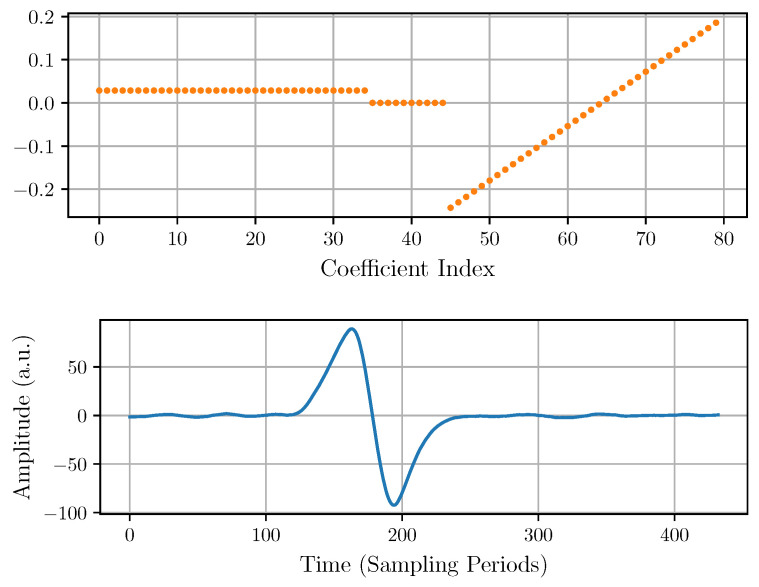
GD FIR coefficients (**top**) and the output pulse corresponding to an experimental input pulse (**bottom**).

**Figure 7 sensors-22-04776-f007:**
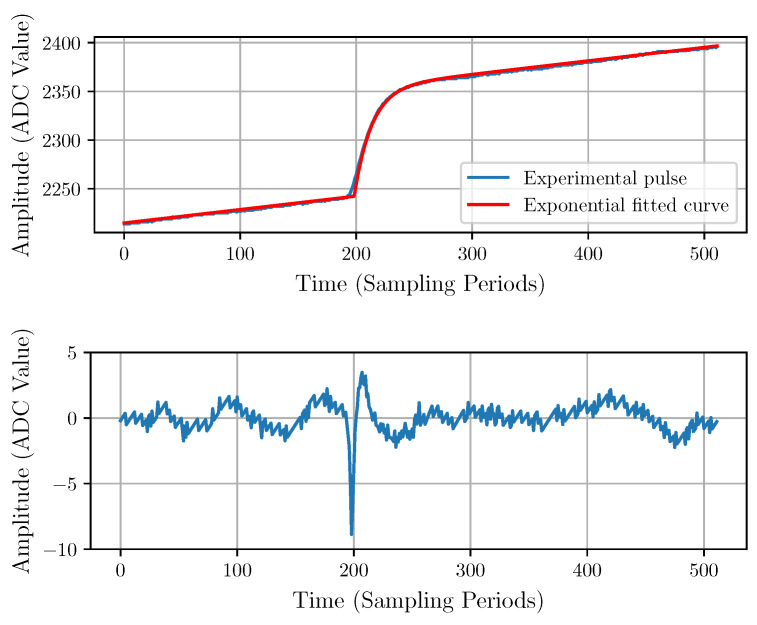
Exponential model fitting (**top**) with its corresponding residuals (**bottom**).

**Figure 8 sensors-22-04776-f008:**
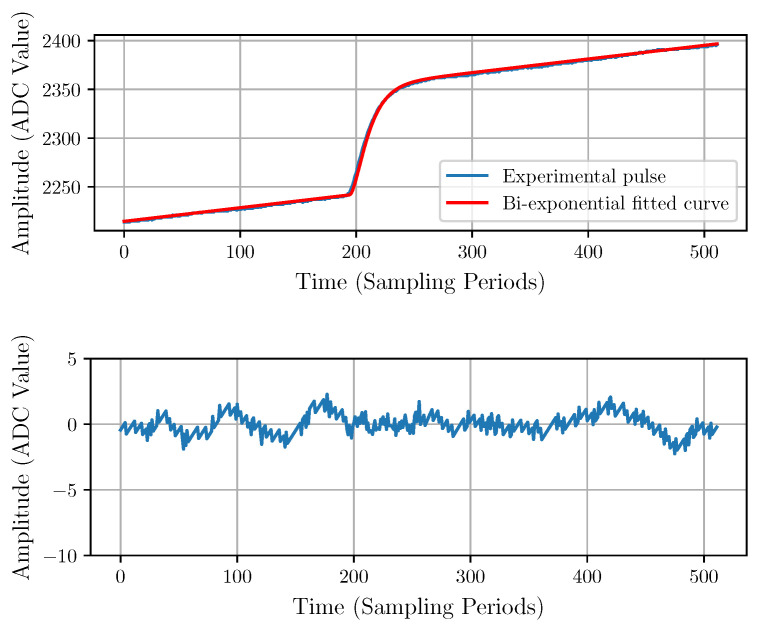
Bi-exponential model fitting (**top**) and corresponding residuals (**bottom**).

**Figure 9 sensors-22-04776-f009:**
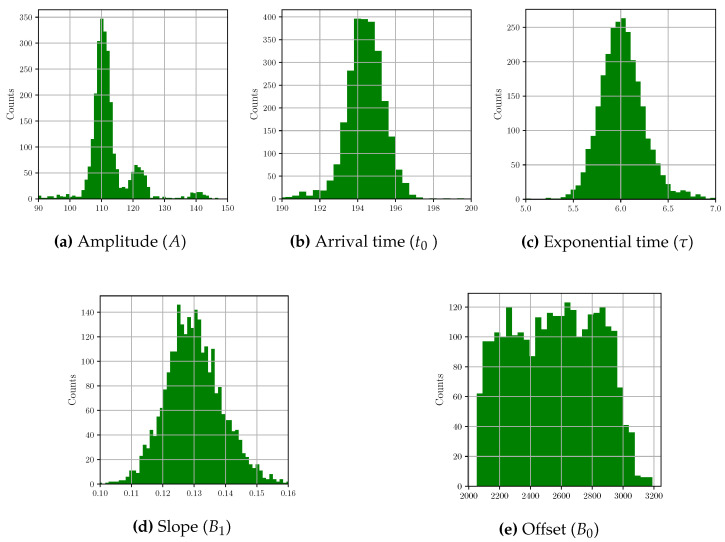
Histograms of the fitted parameters corresponding to the bi-exponential model.

**Figure 10 sensors-22-04776-f010:**
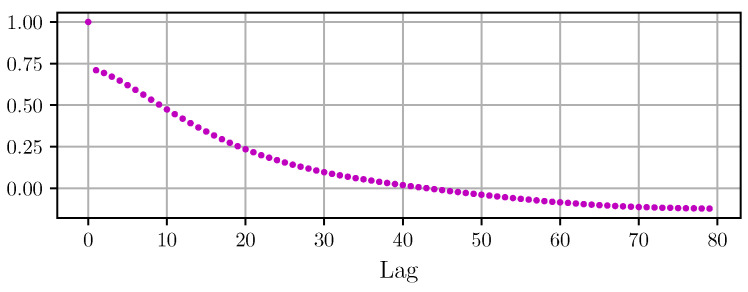
Normalized average autocorrelation function estimated from the residuals of the fitted photon segments.

**Figure 11 sensors-22-04776-f011:**
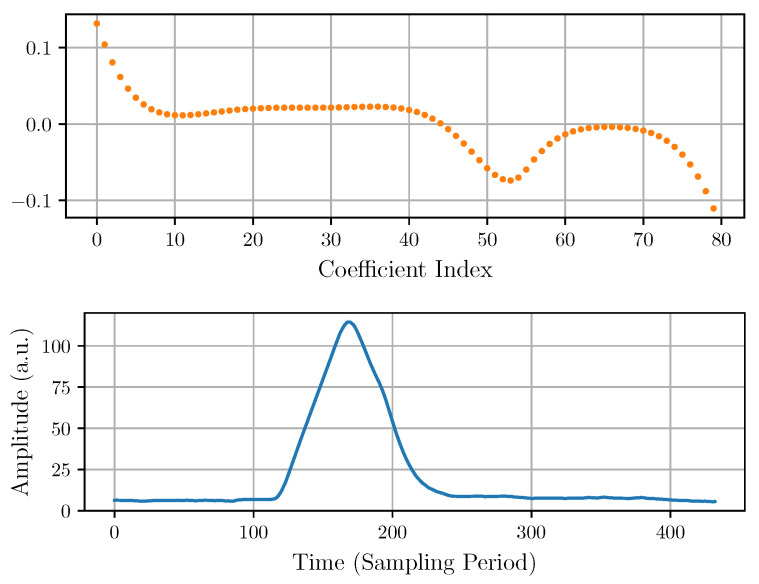
DPLMS FIR coefficients (**top**) and the corresponding output after being applied to an experimental pulse (**bottom**).

**Figure 12 sensors-22-04776-f012:**
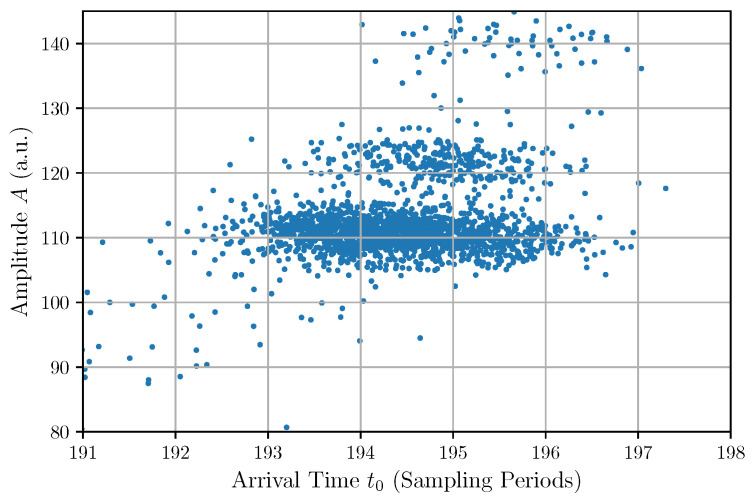
Scatter plot of the amplitude vs. the arrival time.

**Figure 13 sensors-22-04776-f013:**
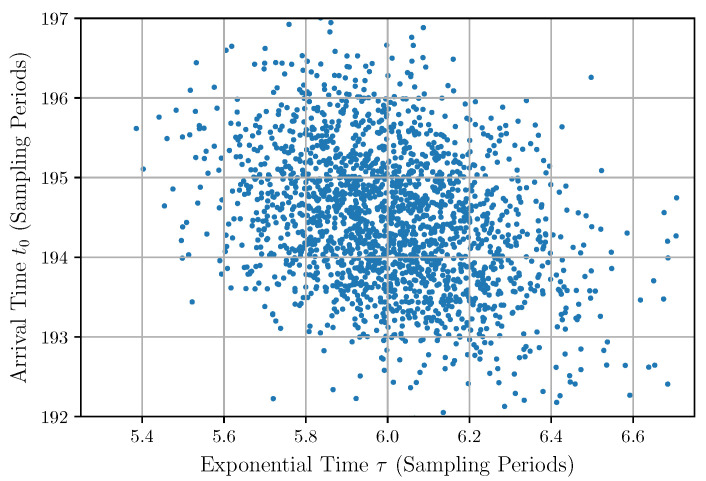
Scatter plot of the arrival time vs. the exponential time.

**Figure 14 sensors-22-04776-f014:**
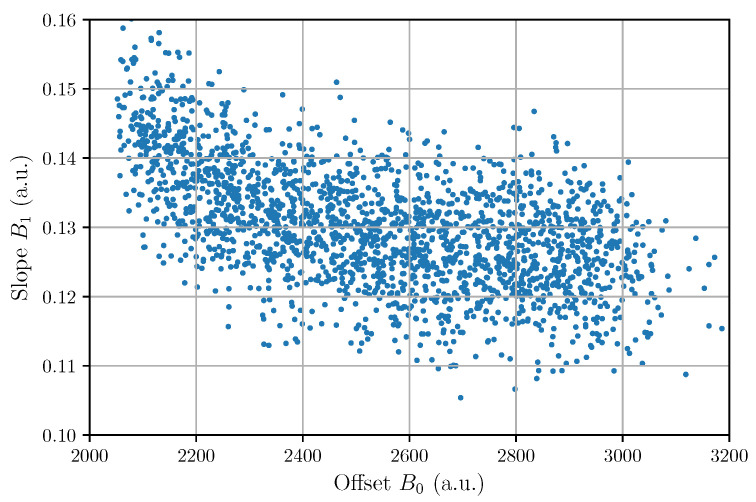
Scatter plot between offset and slope.

**Figure 15 sensors-22-04776-f015:**
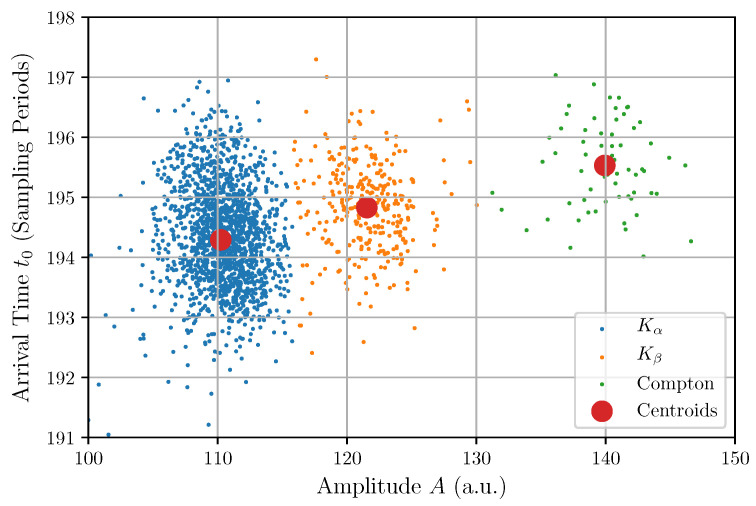
Scatter plot of amplitudes and arrival times with colored clusters and corresponding centroids.

**Table 1 sensors-22-04776-t001:** Pulse models comparison.

	Exponential Model	Bi-Exponential Model
**Mean quadratic residuals **	6201	5914
**Mean peak-to-peak residuals**	13.7	6.6
**Mean Akaike information criterion**	1720	1397

**Table 2 sensors-22-04776-t002:** Mean values and standard deviations of the fitted model parameters.

Parameter	Mean Value	Standard Deviation
**Slope ($B_{1}$)**	0.13	0.01
**Arrival time ($t_{0}$)**	194.27	1.42
**Exponential time (τ)**	6.02	0.24
**Offset ($B_{0}$)**	2547.90	272.29

**Table 3 sensors-22-04776-t003:** Comparison of energy resolutions with different methods to estimate the energy spectrum.

Method	FWHM $K_{\alpha }$ [eV]	FWHM $K_{\beta }$ [eV]	Slope-Error Correction
**GD FIR**	286±4	316±16	yes
**Fitting** ${}^{†}$	267±4	288±17	yes
**Trapezoidal FIR**	207±3	247±17	no
**DPLMS FIR**	202±2	233±12	no

^†^ These results correspond to the histogram of the amplitudes obtained by fitting all available photon traces.

**Table 4 sensors-22-04776-t004:** Correlation distance $C_{D}$ between fitted parameters.

	Arrival Time	Amplitude	Exponential Time	Offset	Slope
**Arrival time**	0	0.53	1.41	0.96	0.96
**Amplitude**	0.53	0	1.05	0.98	1.09
**Exp. times**	1.41	1.05	0	1.02	1.16
**Offset**	0.96	0.98	1.02	0	1.54
**Slope**	0.96	1.09	1.16	1.54	0

## Data Availability

The data that support the findings of this study are available from the corresponding author upon reasonable request.

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
