# Peer review of "Data Analysis and Filter Optimization for Pulse-Amplitude Measurement: A Case Study on High-Resolution X-ray Spectroscopy"

_sensors, 2022, doi:10.3390/s22134776_

Round 1
Reviewer 1 Report
The authors presented a procedure to optimize a finite impulse response shaping filter for digital pulse processing. An adapted digital penalized least mean square (DPLMS) method was employed, which required an comprehensive analysis of experimental data to formulate an accurate mathematical model of the pulse as well as to characterize the specific noise contained in the data. The authors demonstrated the effectiveness of the procedure. The study of Mn X-ray energy spectrum revealed a significant energy resolution improvement of the kα and kβ lines.
The authors addresses the scientific problem nicely and provide sufficient data to support the observation. I recommend to publish.
Author Response
20/06/2022
Dear Reviewer,
Many thanks for your time and for having recommended our manuscript for publication.
We also thank you so much for your positive appreciation.
As suggested, an English writing and style revision has been done.
Sincerely,
Bruno Valinoti, on behalf of all the authors.
Reviewer 2 Report
The authors presented a procedure to optimize a finite impulse response shaping filter using an adapted digital penalized least mean square (DPLMS) method. And the effectiveness of the procedure is demonstrated using a dataset from a case study on high-resolution X-ray spectroscopy based on single-photon detection and energy measurements by formulating accurate mathematical models and characterizing specific data noise.
The manuscript is clearly organized and the experimental data is detailed, but there are still some parts that need to be improved:
1. In the related works section, the time span of many references is relatively long. It is recommended to add more references from the past five years.
2. The data in the tables of the text lack citations. For example, where do the data in Table 3 come from? In Table 3, the difference between the parameters of Trapezoidal FIR and DPLMS FIR methods is not obvious. What are the main advantages of the DPLMS method proposed in this paper?
3. "An improvement of about 20% in the energy resolution" has been pointed out in the abstract and conclusion section, please explain this in detail in the manuscript.
4. A misstatement on line 384, page 15. It should be figure15.
5. There are inconsistencies in the format of formulas and references, and further revisions are recommended.
Author Response
20/06/2022
Dear Reviewer,
Many thanks for your time and for having considered our manuscript.
We also thank you so much for your positive appreciation, pertinent comments, and constructive observations for improving the article.
As suggested, an English writing and style revision has been done.
Here are the responses (in red) to each raised point (in blue):
In the related works section, the time span of many references is relatively long. It is recommended to add more references from the past five years.
Three additional references to newer articles were included in the manuscript [4, 31, 35]:
- Abbene, L.; Principato, F.; Gerardi, G.; Bettelli, M.; Seller, P.; Veale, M.C.; Zambelli, N.; Benassi, G.; Zappettini, A. Digital fast pulse shape and height analysis on cadmium–zinc–telluride arrays for high-flux energy-resolved X-ray imaging. Journal of Synchrotron Radiation 2018, 25, 257–271. doi:10.1107/S1600577517015697.
- Walewski, W.; Nowak Vel Nowakowski, P.; Makowski, D. Giga-sample Pulse Acquisition and Digital Processing for Photomultiplier Detectors. Journal of Fusion Energy 2022, 41. doi:10.1007/s10894-022-00320-0.
- de Cesare, C.; Brambilla, A.; Ouvrier-Buffet, P.; Stanchina, S.; Rossetto, O.; Verger, L. An FPGA-based algorithm to correct the instability of high-resolution and high-flux X-ray spectroscopic imaging detectors. Journal of Instrumentation 2018, 13, P08022–P08022. doi:10.1088/1748-0221/13/08/p08022.
The data in the tables of the text lack citations. For example, where do the data in Table 3 come from? In Table 3, the difference between the parameters of Trapezoidal FIR and DPLMS FIR methods is not obvious. What are the main advantages of the DPLMS method proposed in this paper.
The data reported in table 3 correspond to four different methods that we have applied to the same experimental dataset. To clarify this, we have modified the first sentence of Section 6 as follows:
“We have described and applied four methods to obtain the energy spectrum from the same experimental dataset of single-photon pulses”
The optimization parameters of the DPLMS are the weights that control the constraints: flat-top, specific noise, offset and background slope. The two main advantages of the DPLMS method proposed in the article are that (i) it is the only method that considers the specific characteristics of the noise, which is completely characterized by its normalized autocorrelation function, and (ii) it allows control of the output pulse top flatness. Moreover, it is possible to provide different degrees of correction of the error due to the background slope. To emphasize this aspect, we have modified the following sentence at the end of section 6:
“This method is the only one that considers the specific noise in the dataset and simultaneously allows control of the top flatness of the output pulse. In order to obtain the best energy resolutions, we have relaxed the weight of the slope error correction.”
"An improvement of about 20% in the energy resolution" has been pointed out in the abstract and conclusion section, please explain this in detail in the manuscript.
We have added the following paragraph at the end of section 6:
The fitting of individual photon pulses is a numerically heavy procedure to obtain the pulse amplitude. Although this method is not suitable for online data processing, it is expected to provide the most precise spectrum. The best results however, were obtained with the DPLMS FIR method, which outperformed the fitting procedure by about 20% in terms of energy resolution.
Regarding this point, we have also changed “values” to “reference values” at the end of the abstract
A misstatement on line 384, page 15. It should be figure15.
Corrected
There are inconsistencies in the format of formulas and references, and further revisions are recommended.
The format of formulas and references has been reviewed and corrected. The reference words “Equation, Figure, and Table” are now capitalized everywhere.
Sincerely,
Bruno Valinoti, on behalf of all the authors.

Reviewer 3 Report
Dear authors,
I would like to ask you a few questions and suggestions:
1. Can the аrchitecture of the proposed communication system be easily implemented in practically any modern programmable system on a chip?
2. Text:
Even though we considered that the background slope is constant, a negative trend can be observed between offsets and slopes in figure 14. This effect is also reflected on the value of the correlation distance between the offset and slope vectors. This value indicates that the gain of the CSA is not constant, which in turn determines a non-linear behaviour By further elaborating the relationship between the amplitude of the photons and their arrival time using a clustering procedure, three clear clusters were found and are shown in figure 14. The two largest clusters (left and center) correspond to the Kα and the Kβ transition lines of the Mn, while the third and smallest cluster (right) can be identified as Compton scattered photons. From the linear least-square fit of the centroids of the clusters (large red points), it is found that the arrival time t0 depends approximately on the amplitude A according to t0 = 189.967 + 0.0395A.
- Is it always like that or not? Is there any other way to boost CSA? Discuss the dependence of the detected arrival time on the pulse amplitude.
3. During application, a compromise must often be found between noise reduction and dead time due to the effects of accumulation. For a FIR filter, the step response length can be easily adjusted by the number of filter leads. Filter taps are weighted by coefficients that define the transmission function. The task when designing optimal filters is, therefore, to find the coefficients that lead to minimal noise at the filter output while meeting the requirements in the transmission function.
- What does this mean for a specific application case?
4. What is the speed of X-ray signals and what is the impact on applications that want to gain spectrum quickly with the use of short filters?
5. Discuss the values shown in Figure 10
6. What do you think about the title of the manuscript: Filter performance optimization for high energy resolution X-ray spectroscopy? The title of the manuscript is quite long and contains expressions that are not needed (Measurement, Data Analysis, Amplitude). That is what was said in the Abstract.
7. I would take this out of the Abstract: The adaptation of the DPLMS method requires an in-depth analysis of experimental data to formulate an accurate mathematical model of the pulse as well as to characterize the specific noise contained in the data
8. In the Abstract, start the first row: In this study, the procedure is presented to obtain a set of optimized FIR filter coefficients for pulse amplitude measurement targeted for high-energy resolution X-ray spectroscopy. Correct the Abstract.
With respect
Author Response
20/06/2022
Dear Reviewer,
Many thanks for your time and for having considered our manuscript.
We also thank you so much for your positive appreciation, pertinent comments, and constructive observations for improving the article.
As suggested, extensive English writing and style revision has been done.
Here are the responses (in red) to each raised point (in blue):
I would like to ask you a few questions and suggestions:
- Can the architecture of the proposed communication system be easily implemented in practically any modern programmable system on a chip?
Yes. It is possible as long as the device provides enough logic resources, which is the case for the vast majority of FPGAs nowadays.
- Text:
Even though we considered that the background slope is constant, a negative trend can be observed between offsets and slopes in figure 14. This effect is also reflected on the value of the correlation distance between the offset and slope vectors. This value indicates that the gain of the CSA is not constant, which in turn determines a non-linear behaviour By further elaborating the relationship between the amplitude of the photons and their arrival time using a clustering procedure, three clear clusters were found and are shown in figure 14. The two largest clusters (left and center) correspond to the Kα and the Kβ transition lines of the Mn, while the third and smallest cluster (right) can be identified as Compton scattered photons. From the linear least-square fit of the centroids of the clusters (large red points), it is found that the arrival time t0 depends approximately on the amplitude A according to t0 = 189.967 + 0.0395A.
- Is it always like that or not? Is there any other way to boost CSA? Discuss the dependence of the detected arrival time on the pulse amplitude.
This is an important question. Although it could be possible to boost the CSA (e.g., by increasing its output driving strength and hence its slew rate), there is still the problem of the charge collection time in the detector. For example, if the X-Ray photon absorption occurs close to the anode in a SDD, then the effective collection time will be short and vice versa. For instance, the constant fraction discrimination (CFD) method for detection of the arrival time is supposed to be independent of the pulse amplitude, but only if the pulse amplification is perfectly linear. To better explain this point, we have changed the sentence in lines 393 and 394 (at the end of the fourth paragraph of Section 7) into this one:
“This value indicates that the gain of the CSA is not constant and reveals a non-linear behavior which in turn determines an error of the arrival time detection.”
In this study, we have mainly focused on pulse amplitude measurement and we only marginally discuss pulse arrival detection. An important problem that in our opinion is not completely solved and deserves further research. Many thanks for your observation on this issue.
- During application, a compromise must often be found between noise reduction and dead time due to the effects of accumulation. For a FIR filter, the step response length can be easily adjusted by the number of filter leads. Filter taps are weighted by coefficients that define the transmission function. The task when designing optimal filters is, therefore, to find the coefficients that lead to minimal noise at the filter output while meeting the requirements in the transmission function.
- What does this mean for a specific application case?
Right, the length of the FIR for high spectral resolution at high input photon rates in X-Rays spectroscopy is crucial since it determines the trade-off between detection efficiency and energy resolution. In general, the best noise filtering characteristics are achieved with relatively long filter: a higher quantity of FIR coefficients means more degrees of freedom to explore and obtain better transfer functions than with short filters.
- What is the speed of X-ray signals and what is the impact on applications that want to gain spectrum quickly with the use of short filters?
Shorter filters often diminish dead times and pile-up effects, and increase time resolution and photon counting efficiency. In this work, the length of the filter is an external specification and fixed. An interesting work that specifically studies filter length is in ref [17. Rettenmeier, F.; Maurer, L. Design of optimum filters for signal processing with silicon drift detectors. X-Ray Spectrometry 2021, 50, 501–513. doi:10.1002/xrs.3227.].
- Discuss the values shown in Figure 10
As requested, it has been added the following paragraph at the end of Subsection 5.2:
“The normalized average autocorrelation function in Figure 10 shows an abrupt change, from 1 with lag = 0, to about 0.7 with lag = 1, and from there it follows a smooth decay to slightly negative values from lag = 40 onward. The first abrupt change would correspond to a white noise component, whereas the smooth decay would correspond to relatively low frequency components of the noise spectrum.”
What do you think about the title of the manuscript: Filter performance optimization for high energy resolution X-ray spectroscopy? The title of the manuscript is quite long and contains expressions that are not needed (Measurement, Data Analysis, Amplitude). That is what was said in the Abstract.
Thanks for this suggestion. Possible titles, including the one you suggest, have been extensively discussed among the authors and we prefer to keep the title as it is since it has obtained the largest consensus.
- I would take this out of the Abstract: The adaptation of the DPLMS method requires an in-depth analysis of experimental data to formulate an accurate mathematical model of the pulse as well as to characterize the specific noise contained in the data
This statement has been removed from the abstract as suggested.
- In the Abstract, start the first row: In this study, the procedure is presented to obtain a set of optimized FIR filter coefficients for pulse amplitude measurement targeted for high-energy resolution X-ray spectroscopy. Correct the Abstract.
The abstract has been corrected:
“In this study, we present a procedure to optimize a set of finite impulse response filter (FIR) coefficients for digital pulse amplitude measurement. Such an optimized filter is designed using an adapted digital penalized least mean square (DPLMS) method. The effectiveness of the procedure is demonstrated using a dataset from a case study on high-resolution X-ray spectroscopy based on single-photon detection and energy measurements. The energy resolutions of the k_alpha and k_beta lines of the Manganese energy spectrum have been improved by approximately 20%, compared to the reference values obtained by fitting individual photon pulses with the corresponding mathematical model.”
Sincerely,
Bruno Valinoti, on behalf of all the authors.
